# A duplex real-time PCR with probe for simultaneous detection of *Geosmithia morbida* and its vector *Pityophthorus juglandis*

**Domenico Rizzo**[1], **Daniele Da Lio**[2], **Linda Bartolini**[1], **Giovanni Cappellini**[1], **Tommaso Bruscoli**[1], **Matteo Bracalini**[3], **Alessandra Benigno**[3], **Chiara Salemi**[2], **Dalia Del Nista**[4], **Antonio Aronadio**[4], **Tiziana Panzavolta**[3]*, **Salvatore Moricca**[3]

1 Laboratory of Phytopathological Diagnostics and Molecular Biology, Plant Protection Service of Tuscany, Pistoia, Italy, 2 Department of Agricultural, Food and Agro-Environmental Sciences, University of Pisa, Pisa, Italy, 3 Department of Agriculture, Food, Environment, and Forestry (DAGRI), Plant Pathology and Entomology Division, University of Florence, Florence, Italy, 4 Plant Protection Service of Tuscany, c/o Interporto Toscano "Amerigo Vespucci", Collesalvetti, Livorno, Italy

* tpanzavolta@unifi.it

**Data Availability Statement:** All relevant data are within the manuscript and its Supporting Information files.

## Abstract

The cultivation of walnuts (*Juglans* sp.) in Europe retains high economic, social, and environmental value. The recent reporting of the Thousand Cankers Disease (TCD) fungus, *Geosmithia morbida*, and of its vector, *Pityophthorus juglandis*, in walnut trees in Italy is alarming the whole of Europe. Although Italy is at present the only foothold of the disease outside North America, given the difficulties inherent in traditional identification of both members of this beetle/fungus complex, a rapid and effective protocol for the early detection and identification of TCD organisms is an absolute priority for Europe. Here we report the development of an effective and sensitive molecular tool based on simplex/duplex qPCR assays for the rapid, accurate and highly specific detection of both the bionectriaceous fungal pathogen and its bark-beetle vector. Our assay performed excellently, detecting minute amounts of target DNA without any non-specific amplification. Detection limits from various and heterogeneous matrices were lower than other reported assays. Our molecular protocol could assist in TCD organism interception at entry points, territory monitoring for the early identification and eradication of outbreaks, delineation of quarantine areas, and tracing back TCD entry and dispersal pathways.

## Introduction

The mitosporic ascomycete *Geosmithia morbida* Kolařík (Hypocreales, Bionectriaceae) [1–5] is the fungus responsible for the Thousand Cankers Disease (TCD) in walnut trees [6–8]. The fungus is vectored by the bark beetle, *Pityophthorus juglandis* Blackman (Coleoptera, Curculionidae, Scolytinae), also knonwn as the Walnut Twig Beetle (WTB), native to Mexico and the southwestern United States (California, Arizona and New Mexico) [6]. This disease owes its name to the high number of coalescent cortical cankers produced by *G. morbida* at the insect

**Funding:** The authors received no specific funding for this work

**Competing interests:** The authors have declared that no competing interests exist.

entrance holes [6]. The recently discovered association between *G. morbida* and *P. juglandis*, recognized as TCD's primary vector, is particularly effective, with the disease being the result of the combined activity of these two organisms. The WTB has not evolved morphological adaptations to transport and preserve the symbiotic fungus; however, emerging from infected host plants, WTB adults carry the conidia of the fungus on their body and then introduce them into healthy walnut trees when they bore through the bark [6, 8]. *G. morbida* invades the tissue just beneath the bark, producing small cankers around the insect's entrance holes. Cankered areas are usually not visible in the early stage of tree decline except by removing a thin layer of external bark [6, 9]. Over time, cankers around WTBs' holes and galleries expand and coalesce, affecting the entire circumference of the branches [6]. Affected trees exhibit yellowing and thinning of foliage, branch dieback and, within some years, die due to the severe fungal infection of the phloem deriving from massive insect infestation [6, 9].

*G. morbida* and *P. juglandis* live in symbiosis on various members of the *Juglans* and *Pterocarya* genera, though the greatest damage occurs on the black walnut (*Juglans nigra* L.) and its hybrids. Other walnut species, such as *J. hindsii* Jeps. ex R.E. Sm. and *J. californica* S. Watson, exhibit a variable degree of susceptibility [10]. The English or Persian nut (*J. regia* L.) appears less susceptible, except when it is grafted onto other more susceptible *Juglans* species. Little is known about the response to infection of *J. microcarpa* Berlandier and *J. cinerea* L., on which careful observations have not yet been made [11, 12]. Another important member of the *Juglandaceae*, *Carya illinoinensis* (Wangenh.) K. Koch, seems resistant to the disease [13].

The causes of the severe dieback of walnut species observed in the United States since the Nineties were at first unclear. Whereas the WTB had been known to occur on native North American walnuts for almost a century [14–17], the harmful association between the ascomycete *G. morbida* and the bark beetle *P. juglandis* became clear in the USA only in 2009 [6]. In the autumn of 2013, both the pathogen and the vector were reported for the first time in Europe on several black walnut plantations at Bressanvido, Thiene and Schio, in the province of Vicenza, Veneto region, Italy [18]. Later, the disease was reported in four other Italian regions, Piedmont, Lombardy, Emilia-Romagna and Tuscany, whereas in a fifth Italian region, Friuli-Venezia-Giulia, only the vector was found [19–21]. A Pest Risk Analysis (PRA) carried out in 2015 scored the risk for a possible pervasive spread of TCD in the European and Mediterranean Plant Protection Organization (EPPO) area as "very high" [22]. This prompted more restrictive EU regulations, which have recently led to the inclusion of both *G. morbida* and its vector *P. juglandis* in the list of the European Union's quarantine pests [23].

TCD constitutes a serious threat to black walnut plantations in the EPPO region, particularly in its southern and eastern parts where extensive plantations were established with this exotic tree species in the past decades with financial support from the EU [23]. But concern also exists that the disease could damage the native *J. regia* (and its hybrids), a species both ecologically and economically important, with both high landscape and monetary value. *J. regia* is, in fact, a main component of the European rural landscape, naturally occurring in the countryside and widely grown as an ornamental or for its valuable wood and fruit, which constitute a profitable market.

The identification of both members of this insect/fungal complex by conventional techniques presents several critical issues. First, the fungus is difficult to isolate with traditional culture-based methods, since it grows very slowly and is easily overgrown in cultures by other fungi inhabiting the walnut's surfaces and inner tissues. These fungi include common contaminants, commensals and endophytes, like those of *Fusarium* sp., whose role has yet to be defined [24]. Another hindrance in conventional identification of *G. morbida* is a certain degree of temperature-dependent pleomorphism exhibited by its colonies, which hampers phenotypic discrimination of *G. morbida* colonies from non-*G. morbida* ones. In addition,

repeated subculturing (2–3 cycles) is necessary to obtain pure cultures [25]. All these investigations require considerable mycological expertise and are both time-consuming and labor-intensive. On the insect side, adult WTB identification also requires expertise; furthermore, immature stages are morphologically indistinguishable from other similar-sized bark beetles [26].

For all the above reasons, the development of molecular methods to detect both members of this insect/fungal complex on infected plant material as early as possible was urgently needed. Hence, we developed diagnostic protocols based on simplex/duplex qPCR assays (hybridization probes) for the simultaneous, rapid and reproducible diagnosis of *G. morbida* and its vector *P. juglandis*. These protocols will aid in the monitoring of walnut plantations, nurseries, gardens, natural areas and ports of entry for the TCD-causing agents. They will also provide valuable support in delineating TCD quarantine zones, as well as in tracing back entry pathways.

## Materials and methods

### Sample collection and TCD organism identification

A survey was carried out in a walnut plantation (composed mainly of *J. nigra* with a few *J. regia*s) in the province of Florence, where an outbreak of TCD had occurred in 2018. Starting from April 2019, a total of 24 symptomatic samples (portions of branches) were randomly collected from infected trees. Samples were checked for the presence of WTB entry/emergence holes on the bark, as well as for galleries under the bark, by peeling off the bark. *P. juglandis* individuals and woody tissues from infested branches were taken to the laboratory.

*G. morbida* was isolated on a Potato Dextrose Agar (PDA, Difco Laboratories, Detroit, MI) nutrient medium; subsequently, the fungus and the WTB were both identified at the morphological and molecular level. According to Moricca *et al.*'s [20] methodology, who worked on samples from the same outbreak site, the rDNA region of *G. morbida* was amplified by end-point PCR, from both pure mycelium and infected plant tissue, via the universal primers ITS4 and ITS6 [27]. A portion of the mitochondrial cytochrome oxidase subunit 1 (COI) gene from *P. juglandis* was amplified by end-point PCR using the primers LCO1490 and HCO2198 [28]. Amplicons of the two organisms were then purified and sequenced. Moricca *et al.* [25] performed a Basic Local Alignment Search Tool (BLAST®) search on the GenBank database for the closest homologies in the ITS and COI sequences of *G. morbida* and *P. juglandis* respectively; they found that the fungus from this outbreak in central Italy (GenBank accession number: MH620784) corresponded 99% to isolates from North America and northern Italy (Veneto). Instead, the beetle (GenBank accession number: MH666050) matched 100% the *P. juglandis* haplotype H1 from the US and Piedmont (north-eastern Italy).

In addition to the target organisms, several non-target organisms (Table 1) were tested to assess the specificity of the diagnostic assay. The non-target organisms were included in the test if they fulfilled at least one of these criteria: a) association with the same host plants; b) taxonomic relatedness; c) endophytic (fungi) or xylophagous (insects) lifestyle; d) frequent interception by the Phytosanitary Service of the Tuscany Region–Italy (SFR) of beetles whose first instars can be confused with *P. juglandis*. All insect samples were stored in 70% ethanol and morphologically identified. When possible, fresh tissue material was used for DNA extraction. Dried specimens (stored at 12–16˚C; 50% relative humidity) provided by the Universities of Florence, Pisa, Bologna, CREA-AA of Florence (Centro di Ricerca Agricoltura e Ambiente) and CNR (Consiglio Nazionale delle Ricerche) were also used. The storage conditions of the specimens obtained from other sources cited in Table 1 are unknown. The nucleic acids of all

**Table 1. List of target and non-target fungi and insects used in this study.**

| Species | Collection date | Source material | Supplier | Host* |
|---|---|---|---|---|
| **Fungi** | | | | |
| *Geosmithia* sp. (Pitt) | 2018 | mycelium | CNR-Florence | - |
| *Geosmithia morbida* (M. Kolařík, Freeland, C. Utley & Tisserat) | 2018 | mycelium | University of Florence | *Juglans nigra* |
| | 2018 | mycelium | SFR Phytopathol. lab | *Juglans nigra* |
| | 2018 | mycelium | CNR-Florence | *Juglans nigra* |
| | 2018 | infected woody tissue | SFR Phytopathol. lab | *Juglans nigra* |
| | 2018 | *P, juglandis* frass | SFR Phytopathol. lab | *Juglans nigra* |
| | 2018 | *P, juglandis* adults | SFR Phytopathol. lab | *Juglans nigra* |
| *Geosmithia obscura* (M. Kolařík, Kubátová & Pažoutová) | 2018 | mycelium | CNR-Florence | - |
| *Geosmithia langdonii* (M. Kolařík, Kubátová & Pažoutová) | 2018 | mycelium | CNR-Florence | - |
| *Botryosphaeria* (Ces. & De Not.) | 2014 | mycelium | SFR Phytopathol. lab | - |
| *Botryosphaeria dothidea* (Moug.) Ces. & De Not. | 2014 | mycelium | University of Bologna | - |
| *Colletotrichum* (Corda) | 2014 | mycelium | SFR Phytopathol. lab | - |
| *Colletotrichum acutatum* (J.H. Simmonds) | 2014 | mycelium | SFR Phytopathol. lab | - |
| *Colletotrichum coccodes* (Wallr.) S. Hughes | 2014 | mycelium | University of Bologna | - |
| *Colletotrichum gloeosporioides* (Penz.) Penz. & Sacc. | 2014 | mycelium | SFR Phytopathol. lab | - |
| | 2018 | mycelium | SFR Phytopathol. lab | *Photinia* sp. |
| *Fusarium oxysporum* (Schltdl.) | 2016 | mycelium | University of Bologna | - |
| *Fusarium oxysporum* f.sp. *radicis-lycopersici* (Jarvis & Shoemaker) | 2015 | mycelium | SFR Phytopathol. lab | *Solanum lycopersicum* |
| *Fusarium redolens* (Wollenw.) | 2016 | mycelium | University of Bologna | - |
| *Gibberella circinata* (Nirenberg & O'Donnell) | 2016 | mycelium | CREA-PAV_Rome | - |
| *Neofusicoccum luteum* (Pennycook & Samuels) Crous, Slippers & A.J.L. Phillips | 2017 | mycelium | University of Florence | *Vitis* sp. |
| *Neofusicoccum parvum* (Pennycook & Samuels) Crous, Slippers & A.J.L. Phillips | 2017 | mycelium | University of Florence | *Vitis* sp. |
| *Neofusicoccum ribis* (Slippers, Crous & M.J. Wingf.) Crous, Slippers & A.J.L. Phillips | 2016 | mycelium | University of Florence | *Vitis* sp. |
| *Neofusicoccum vitifusiforme* (Van Niekerk & Crous) Crous, Slippers & A.J.L. Phillips | 2016 | mycelium | University of Florence | *Vitis* sp. |
| **Insects** | | | | |
| *Pityophthorus juglandis* Blackman | 2018 | Frass | University of Florence | *Juglans nigra* |
| | 2018 | Adult | | |
| *Pityophthorus pubescens* (Marsham) | 2018 | Adult | University of Florence | - |
| *Cossus cossus* (Linnaeus) | 2019 | Frass | University of Florence | - |
| *Euzophera semifuneralis* (Walker) | 2018 | Larva | SFR Phytopathol. lab | *Acer* sp. |
| *Hylurgus ligniperda* (Fabricius) | 2018 | Adult | University of Florence | *Aesculus hippocastanum* |
| *Ips sexdentatus* (Börner) | 2018 | Adult | University of Florence | - |
| *Ips typographus* (Linnaeus) | 2014 | Adult | SFR Phytopathol. lab | *Acer* sp. |
| *Orthotomicus erosus* (Wollaston) | 2018 | Adult | University of Florence | - |
| *Sesia* sp. | 2019 | Frass | University of Naples | *Prunus sp.* |
| *Tomicus destruens* (Wollaston) | 2018 | Adult | University of Florence | - |
| *Xyleborus dispar* (Fabricius) | 2020 | Adult | CREA_AA_Florence | - |

(*Continued*)

**Table 1.** (*Continued*)

| Species | Collection date | Source material | Supplier | Host* |
|---|---|---|---|---|
| *Xyleborus monographus* (Fabricius) | 2020 | Adult | CREA_AA_Florence | - |
| *Xyleborinus saxesenii* (Ratzeburg) | 2018 | Adult | University of Florence | - |
| | 2020 | Adult | CREA_AA_Florence | - |
| *Xylosandrus compactus* (Eichhoff) | 2018 | Adult | SFR Phytopathol. lab | *Laurus nobilis* |
| *Xylosandrus crassiusculus* (Motschulsky) | 2018 | Adult | University of Pisa | *Malus* sp. |
| | 2019 | Frass | | |
| *Xylosandrus germanus* (Blandford) | 2019 | Adult | University of Florence | - |
| *Zeuzera pyrina* (Linnaeus) | 2019 | Frass | SFR Phytopathol. lab | *Olea europea* |

*Host plants from which organisms were collected.

the processed fungi were extracted and stored at -20°C, then species-specific diagnostic protocols were applied.

## DNA extraction

The nucleic acids of the target organisms were extracted from four different types of matrices: symptomatic wood samples; insect frass; insect specimens; and fungal mycelia. For all of these, a single extraction method was used: 2% CTAB [29], with some modifications depending on the type of matrix (Fig 1). Specifically, the homogenization phase was diversified as follows: for symptomatic wood tissues and insect frass about 1 g of matrix was homogenized by means of 10 mL steel jars using a TissueLyzer (Qiagen) for 30 seconds at 3000 oscillations per minute; for fungal mycelia about 100 mg were macerated into Eppendorf microtubes (Sarstedt, Germany) by using micro pestles; each insect specimen was ground and homogenized individually using nylon mesh U-shaped bags (Bioreba, Reinach, Switzerland). Variable volumes of 2% CTAB buffer (2% CTAB, 1% PVP-40, 100 mM Tris–HCl, pH 8.0, 1.4 M NaCl, 20 mM EDTA, and 1% sodium metabisulfite) were added immediately: 7 mL for woody tissues/insect frass and 1 mL for mycelia and insects. 0.5–1 mL of lysate was then incubated at 65°C for 10 minutes. 1 volume of Chloroform was added, stirred by inversion and centrifuged at 13,000 rpm for 5 minutes. 600 μL were taken from the supernatant and an equal volume of isopropanol was inserted, mixed by inversion and also centrifuged at 13,000 rpm for 5 minutes. The resulting pellet was dried by speed vacuum for 10 minutes, then resuspended in 100 μL of sterile, ultra-pure water and incubated at 65°C for 5 minutes.

DNA was extracted from 24 symptomatic wood samples, 6 frass samples, 16 *G. morbida* mycelial samples and 12 adult insects taken from the symptomatic wood samples. Each type of sample (woody pieces, frass, fungal mycelium and insects) was extracted in duplicate. Quantization, contamination degree, and evaluation of the extracted DNA were performed using the QIAxpert (Qiagen). The DNA was eluted in 100 μL of nuclease-free water and either used in real-time PCR immediately or stored at -20°C until use. To check the performance of the DNA extracted from the insects, 1:20 DNA/ddsH$_2$O was tested in a real-time PCR using a dual-labeled probe targeting a highly conserved region of the 18S rDNA [30]. In addition, further tests of differentiated amplificability were carried out depending on the matrix (insects and wood samples). In fact, for infected/symptomatic wood samples, the quality and integrity of the extracted DNA were assessed by verifying amplificability through a real-time PCR with a dual-labeled probe for the plant COI gene [31]. These amplificability tests served as quality

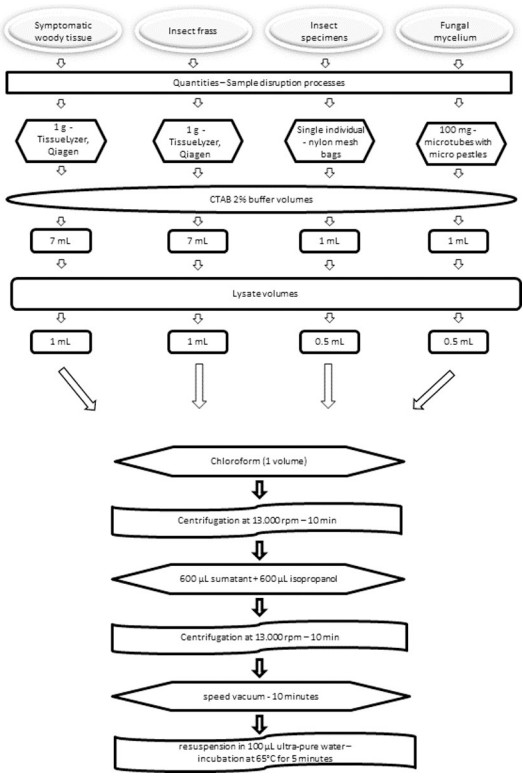

**Fig 1. Flow chart of the major steps of the DNA extraction method from the four different matrices.** It takes 50 minutes hand-on time to process 48 samples. One technician can process 144 samples in a working day.

controls for the extractions. They also allowed us to look for inhibitors in the wood pieces by calculating the Cq and the slope of the relative amplification curves.

## Design of primers and probes and their optimization

Primer pairs and probes were designed for *G. morbida* and *P. juglandis* using the OligoArchitect™ Primers and Probe Online software (Sigma-Aldrich, St. Louis, USA) with the following specifications: 80 to 200 bp product size; Tm (melting temperature) 55 to 65° C; primer length 18 to 26 bp; and absence of secondary structures whenever possible (Table 2). The beta-tubulin and COI genes were utilised for designing the oligos and probes of, respectively, *G. morbida* and *P. juglandis*. An *in-silico* test of the primer pairs was then performed with the BLAST®

**Table 2. Primers and probes for protocols in qPCR probes for TCD organisms.**

| Primer pair and probe | Length (bp) | Sequence | Position | Product size (bp) | Sequence |
|---|---|---|---|---|---|
| | | *Geosmithia morbida* | | | |
| Gmorb_228_F | 18 | GGAGATGGCGTCTCTTTG | 228 to 246 | | |
| Gmorb_319_R | 18 | ACGAGAGTCAGTGTTCTG | 319 to 301 | 92 bp | KJ148218.1 |
| Gmorb_255_P | 23 | FAM—TCTACCTCTTCCTGTCCAGCCTA—BHQ1 | 255 to 278 | | |
| | | *Pityophthorusjuglandis* | | | |
| Pjug_253_F | 22 | TCCCACGTCTTAATAATATAAG | 253 to 275 | | KX809936.1 |
| Pjug_435_R | 20 | CTCCTGCTATATGAAGACTA | 435 to 415 | 183 bp | |
| Pjug_281_P | 27 | Hex_ACTCTTACCACCATCATTAACATTCCT_BHQ1 | 281 to 308 | | |

software to assess the specificity of the designed primer pairs. As further verification of the *in silico* specificity, the most-related nucleotide sequences found by the BLAST software (the expected amplicons of the probe-based real-time PCR protocol were queried), were aligned using the MAFFT program [32] (Figs 2–5), implemented within the Geneious® 10.2.6 software (Biomatters, http://www.geneious.com).

For both *G. morbida* and *P. juglandis*, thermal gradients were made from 52˚C to 60˚C, to determine the optimal annealing temperatures. The concentration of oligos and probes was also evaluated, carrying out checks at various concentrations for both: 0.1 µM, 0.2 µM, 0.3 µM and 0.4 µM. The amplification reactions in the qPCR probe were performed with a CFX96 (Biorad) thermocycler at a final volume of 20 µl. The DNA samples were amplified in 96 wells of 0.2 mL plates for real-time PCR (Starlab, Milan). Each reaction was carried out twice. For each real time PCR, two tubes with 2 µL dd-water were used as No Template Controls (NTC). Positive amplification controls for each target organism and a negative amplification control was included in each real time PCR run. Samples were tested as technical duplicates and tests were repeated when results appeared unclear or contradictory. Bio-Rad CFX96 data were analyzed with CFX Maestro 1.0 software using automatic thresholds and baselines for FAM and HEX.

After initially optimizing reactions for each qPCR simplex probe, both in terms of concentrations and in terms of execution temperatures, we searched for the best combination for the duplex qPCR with the two pairs of primers and the two probes used simultaneously. Then the thermal gradient operations were repeated from 52˚C to 60˚C (annealing).

## Performance characteristics of the qPCR Probe

Analytical sensitivity and specificity, repeatability and reproducibility were the performance criteria we used to evaluate the usability of these tests for routine diagnostics. Validation was performed according to EPPO standard PM7/98, 2019 (4). The analytical sensitivity (the limit of detection, or LoD) for the real-time probe protocols for *G. morbida* and *P. juglandis* was evaluated using a 10-fold 1:5 serial dilution in triplicate from genomic DNA extracts (10 ng/ µL) from infected woody tissue and from single adult insects respectively. The diagnostic specificity of the real-time PCR assay was tested using extracted genomic DNA, at a final

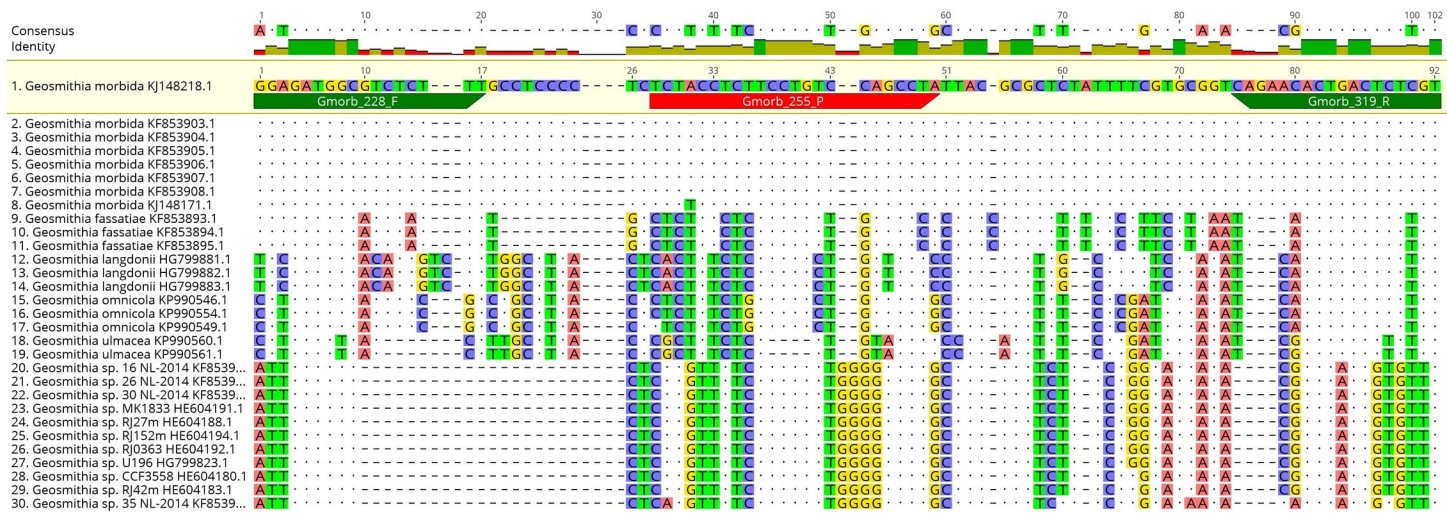

**Fig 2. Partial sequence alignment of the beta-tubulin gene from the target *G. morbida* and homologous sequences of related *Geosmithia* sp. in GenBank using the probe real-time PCR protocol similarity values.**

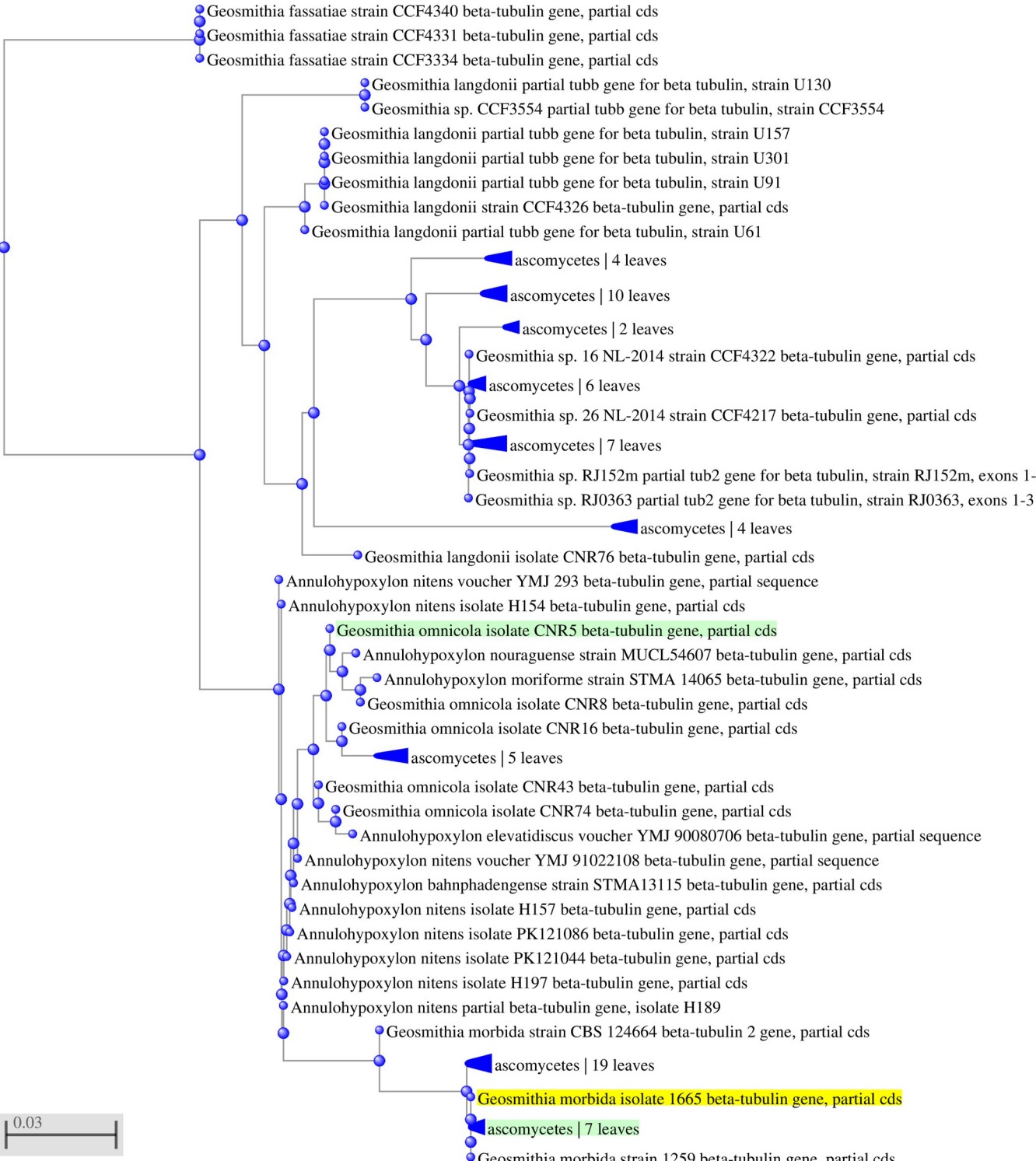

**Fig 3. Unrooted phylogenetic tree from the GenBank sequences of *G. morbida* and related (non-*G. morbida*) species using the probe-based qPCR protocol.** The phylogenetic tree was constructed using Geneious 10.2.4 according to the neighbor-joining method and the Tamura-Nei model with 1000 bootstrap replicates.

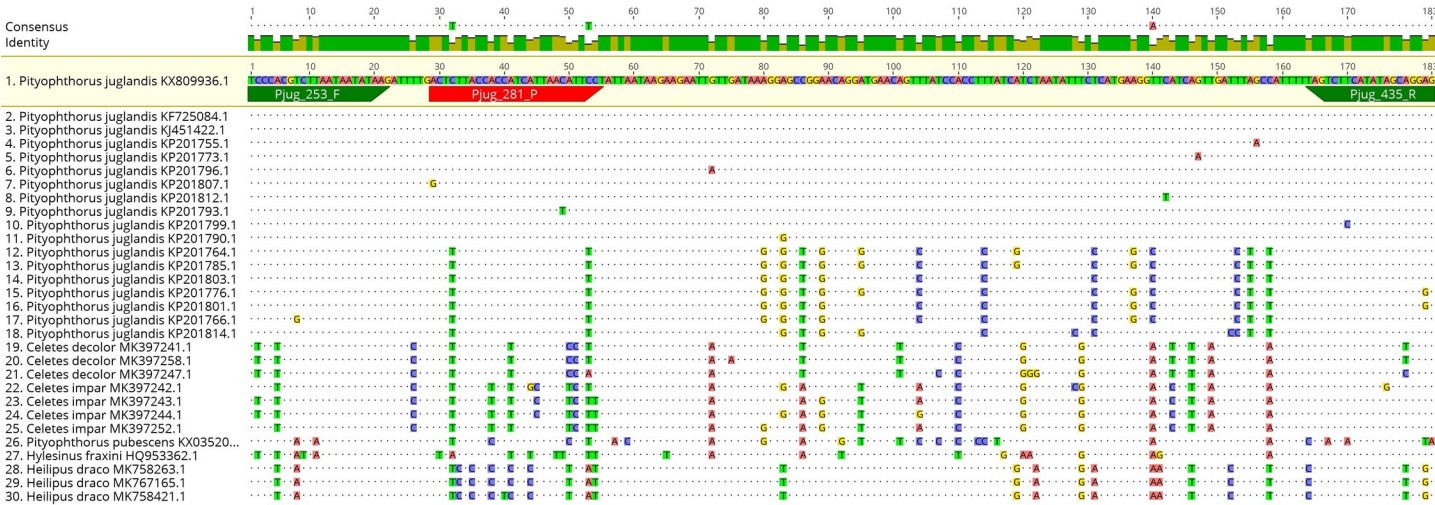

**Fig 4. Partial sequence alignment of the mitochondrial cytochrome oxidase subunit 1 (COI) gene from the target *P. juglandis* and homologous sequences of related insects in GenBank using the probe real-time PCR protocol similarity values.**

concentration of 10 ng/µL, from target and non-target organisms/source samples. The evaluation range for all protocols studied was from 10 ng/µL to 25.6 fg/µL. All measurements were made using the QIAxpert (Qiagen); as far as possible, the same serial dilutions were used to compare the various techniques. Mean Cq values and standard deviations (SDs) were calculated for the target species. The diagnostic protocols were tested (to evaluate the repeatability of the assay) on 10 samples of *P. juglandis* adults and *G. morbida* isolates, diluted at a concentration of 5 ng/µl, with three independent extractions performed on each sample. The reproducibility for each developed protocol was the same as that for repeatability but with two different operators and on different days. The raw data of qPCR amplification were analyzed using the CFX Maestro 1.0 software (Biorad). The DNA concentration, OD ratio and repeatability/reproducibility results were analyzed statistically using descriptive parameters (percentage coefficient of variation and standard deviation) via SPSS version 26.0 (SPSS Inc., Chicago, IL).

## Results

### DNA extraction

The effectiveness of the DNA extraction can be deduced from Table 3, where the average values of the concentrations (ng/µl) of DNA extracted from the various, heterogenous matrices, their standard deviation averages (SD) and the absorbance averages (260/280) are reported. The real time PCRs for the COI gene [31] gave an average Cq value of 18.56, with a standard deviation of 1.36. Moreover, the DNA amplificability, assessed on adult specimens of *P. juglandis*, revealed DNA extracts were perfectly amplifiable with the real-time PCR probe targeting the 18S rDNA Universal [30], with an average Cq value of 15.64 (± 1.86 SD).

### Optimization of the probe-based real-time PCR assay conditions for singleplex e duplex

The optimal reaction mix ended up being 10 µL of 2x QuantiNova Probe PCR Master Mix (Qiagen) with 0.4 µM of primers and a 0.2 µM probe concentration. The optimal annealing temperature for the qPCR probe (simplex) reactions was 60˚C for *G. morbida* and 55˚C for *P.*

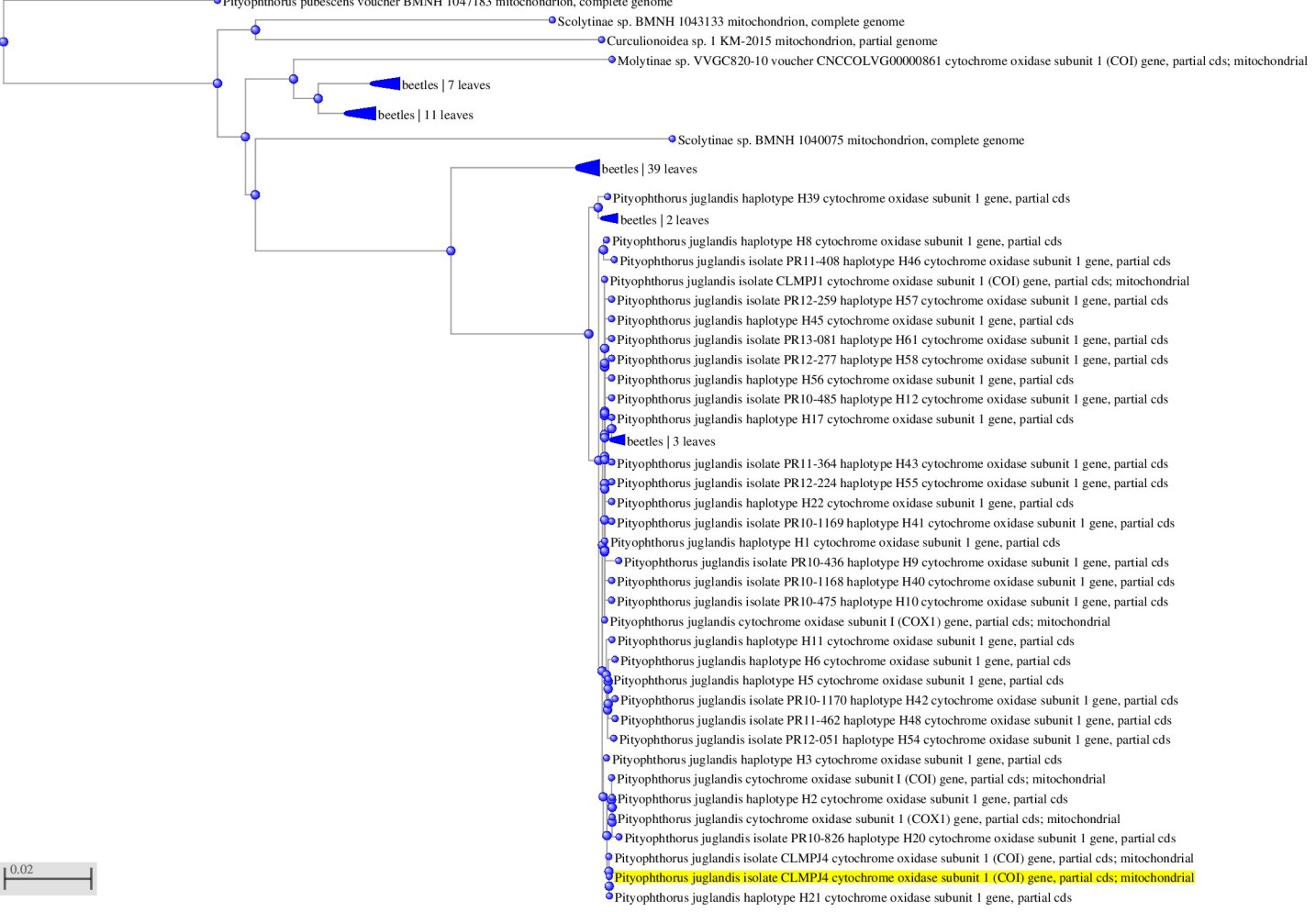

**Fig 5. Unrooted phylogenetic tree from GenBank sequences of *P. juglandis* and related (non-*P. juglandis*) species used in the probe-based qPCR protocol.** The phylogenetic tree was constructed using Geneious 10.2.4 according to the neighbor-joining method and the Tamura-Nei model with 1000 bootstrap replicates.

*juglandis*. There was no significant difference in the Cq values for any of the *G. morbida* or *P. juglandis* DNA samples when the assay was run with annealing temperatures of 52, 54, 56, 58, 60 and 62°C. Similarly, there were only negligible differences in Cq values among the different

**Table 3. DNA concentrations obtained from the four different matrices.**

| CTAB 2% extraction method (from Li *et al*. 2008 [29], modified) | | |
|---|---|---|
| **Type of matrix** | **DNA concentration** | **A$_{260/280}$** |
| | **ng/µl (mean values) ± SD** | **(mean values)** |
| Infected/symptomatic woody tissue | 85 ± 6.8 | 1.8 ± 0.2 |
| Adult *P. Juglandis* insects | 102 ± 10.6 | 1.9 ± 0.16 |
| *G. morbida* fungal mycelium | 68 ± 5.8 | 1.7 ± 0.21 |
| Insect frass | 26 ± 4.2 | 1.68 ± 0.2 |

Mean values of DNA concentrations with standard deviation (SD) and absorbance (A) obtained with extractions from the four different matrices.

concentrations of primers (300 and 500 nM) or probe (150 and 250 nM). These data confirm the robustness of the assay. The duplex real-time PCR assay was then optimized starting with the above primer and probe concentrations, which had been optimized for the simplex q-PCR probe, and by setting a gradient that reconciled the two optimal annealing temperatures for the two target organisms: 95˚C for 2 min followed by 95˚C for 10 seconds and annealing at 58˚C for 40 seconds. Samples were considered positive, for all duplex q-Probe reactions, when the resulting real-time PCR curves showed an evident inflection point (in addition to increasing kinetics) and Cq values <35.

## Performance characteristics

All the assays were inclusive for *G. morbida* and *P. juglandis* and exclusive for the non-target organisms tested. *G. morbida* was detected from all samples taken from symptomatic woody tissues, both in simplex and in duplex modes, as well as from the fungal mycelium. In addition, traces of the fungus were found in 5 of the 12 adults of *P. juglandis*. Furthermore, the assays were effective in detecting the presence of *P. juglandis*, both in simplex and in duplex modes, from the insect frass found in tree branches, even if with a borderline Cq (about 35).

The diagnostic specificity was therefore 100% for all the protocols we developed (simplex). The verification of the diagnostic specificity [21, 33] was 100%, too. All target specimens were correctly identified using the specific tests and no false positives (within the reference cut-off values for qPCR assays) were obtained with non-target organisms, resulting in 100% diagnostic sensitivity and specificity for all the tests. All test runs yielded the same qualitative results and were not influenced by variation in testing conditions. The validation parameters examined were also applied to the duplex real-time PCRs. Again, the specificity tests carried out on the target and non-target organisms revealed the method was 100% inclusive, specific and diagnostically sensitive. The correlation levels were very good with $R^2$ equal to 0.98 and 0.99 for *G. morbida* and *P. juglandis* respectively. Likewise, the slope of the standard curve was equal to 3.06 and 3.13 for *P. juglandis* and *G. morbida*, respectively. The analytical sensitivity (LoD) evaluated for the q-Probe protocols (10-fold 1:5 serial dilutions in triplicate) for *G. morbida* and *P. juglandis* was equal to 3.2 pg/µL and 25.6 fg/µL, respectively (Table 4).

The repeatability and reproducibility gave very low values in both protocols (Table 5). In the qPCR probe protocol for *G. morbida*, the repeatability intervals were between 2.83 and

**Table 4. Analitycal sensitivity (LoD) assays.**

| Dilutions 1:5 | qPCR probe *G. morbida* | qPCR probe *P. juglandis* |
|:---:|:---:|:---:|
| | Cq means ± SD | Cq means ± SD |
| 10 ng/µL | 25.54 ± 0.48 | 18.31 ± 1.15 |
| 2.0 ng/µL | 27.52 ± 0.43 | 20.69 ± 0.67 |
| 0.4 ng/µL | 29.57 ± 0.47 | 23.05 ± 0.34 |
| 0.08 ng/µL | 31.12 ± 0.14 | 24.57 ± 0.21 |
| 0.016 ng/µL | 33.05 ± 0.53 | 26.85 ± 0.47 |
| 3.2 pg/µL | 34.75 ± 0.19 | 28.35 ± 0.43 |
| 0.64 pg/µL | - | 30.16 ± 0.17 |
| 0.128 pg/µL | - | 32.30 ± 0.05 |
| 25.6 fg/µL | - | 33.53 ± 0.64 |

Analitycal sensitivity (LoD) assays using 1:5 serial dilutions (from 10 ng/µL to 25.6 fg/µL). Mean Cq±SD = mean of the three threshold cycles of each dilution (Cq) ± standard deviation (SD). Cq values above 35 were considered as negative results.

**Table 5. Repeatability and reproducibility of real-time assays computed as a percent coefficient of variation (% CV).**

| Sample | qPCR probe protocol: *G. morbida* | | | qPCR probe protocol: *P. Juglandis* | | |
|---|---|---|---|---|---|---|
| | Repeatability (%CV) | | Reproducibility | Repeatability (%CV) | | Reproducibility |
| | Assay 1 | Assay 2 | (%CV) | Assay 1 | Assay 2 | (%CV) |
| 1 | 0.28 | 0.77 | 0.40 | 2.20 | 2.23 | 0.43 |
| 2 | 0.21 | 0.05 | 0.00 | 5.06 | 0.52 | 1.46 |
| 3 | 0.64 | 0.50 | 0.66 | 4.50 | 0.21 | 0.38 |
| 4 | 0.19 | 0.73 | 0.36 | 2.50 | 0.99 | 1.48 |
| 5 | 0.00 | 0.28 | 0.45 | 3.88 | 1.32 | 0.86 |
| 6 | 0.17 | 0.97 | 0.14 | 0.29 | 0.47 | 2.50 |
| 7 | 0.65 | 2.83 | 0.65 | 0.32 | 3.25 | 1.65 |
| 8 | 1.92 | 0.24 | 1.38 | 2.81 | 1.14 | 1.41 |
| 9 | 0.51 | 0.41 | 0.36 | 4.63 | 1.59 | 4.36 |
| 10 | 0.39 | 0.51 | 0.46 | 0.76 | 2.99 | 0.55 |

0.05, while for the reproducibility they were between 1.38 and 0. As regards the qPCR for *P. juglandis*, the repeatability limit values were 5.06 and 0.21, while the reproducibility ranged from 4.36 to 0.38.

## Discussion

The diagnosis of the two TCD-causing organisms by traditional means is not straightforward, being hindered by several drawbacks inherent in the initiation of *G. morbida* mycelial growth on nutrient media, the subsequent ontogeny of its colonies, and the developmental biology of WTB. *G. morbida* grows very slowly on artificial laboratory media and this causes it to be out-competed in the culture by other fungal co-inhabitants of the inner tissue of walnut trees. Specifically, these endophytic fungi grow faster than the agent of TCD and it is not uncommon that they completely overgrow *G. morbida*, thus masking its presence [24]. In fact, a *G. morbida* colony takes from 2 to 5 weeks to go from initial mycelial networking to subculturing in axenic culture. Moreover, colony phenotypes are often unstable following repeated subculturing or as the growth temperature changes. Colonies are normally lobed, but their derivatives often exhibit intra-isolate lobe variation, thus with changes in the type of margin, as well as variation in mycelium texture, compactness and pigmentation. The bark beetle, on the other hand, is easily confused with similar-sized beetles, as at preimaginal stages it lacks distinctive diagnostic traits [26]. Morphological identification of both TCD organisms is thus technically challenging, requiring an expert staff, as well as time-consuming, especially when a high number of samples must be processed [17]. However, efficient and high-performing molecular diagnostic methods have been developed in recent years for the detection of several quarantine organisms [34–36]. These methods are rapid, sensitive and specific, overcoming the time-consuming and often troublesome traditional identification. Here we applied them to the problem of diagnosing *G. morbida* and its main vector *P. juglandis*, thereby developing a sensitive and highly performing, species-specific, molecular diagnostic tool.

A further difficulty in diagnosing this beetle/fungus complex is the need to check different matrices which might contain the fungus (e.g. the mycelium and propagules; plant tissue) or the beetle (e.g. adults, larvae, frass and plant tissue). For this reason, our initial step was to develop a well-performing and reliable DNA extraction protocol that could be used for various source materials. Indeed, our extraction method is both fast and versatile, providing good results from a large variety of matrices from both insects and fungi, and with good

performance for large scale phytosanitary investigations and diagnostic screenings. Specifi-
cally, our DNA extraction protocol allows the processing of up to 24 single-insect samples or
wood tissues in a relatively short time (about 50 minutes). This protocol is quick to perform,
taking no more than two hours from DNA extraction to analysis of real-time PCR results.
Amplificability tests carried out to control the effectiveness of the extractions verified the
absence of inhibitors either to the detected Cq or to the slope of the relative amplification
curves. Furthermore, amplificability checks with both insects and woody matrices gave very
good results.

The validation parameters, such as specificity, sensitivity and diagnostic accuracy, provided
excellent values. The individual LoDs of the two methods gave sound results, too. The qPCR
probe test from *P. juglandis* adults (25.6 fg/μL) showed a greater analytical sensitivity com-
pared to that from symptomatic wood tissue infected with *G. morbida* (3.2 pg/μL), an outcome
that probably depended on the different nature of the source matrices. Furthermore, the
repeatability and reproducibility values for both *G. morbida* and *P. juglandis* suggested low
intra-run and inter-run mean variation. In fact, maximum values were 2.83 and 1.83 (for
repeatability and reproducibility, respectively) for *G. morbida*, while slightly higher values,
5.06 and 4.36, were observed for *P. juglandis*, probably to be ascribed to the troublesome start-
ing matrix (adult insects).

Other protocols have been developed for the molecular diagnosis of TCD by Lamarche
*et al*. [37], Oren *et al*. [24], and Moore *et al*. [15]. These methods involved looking for *G. mor-
bida* on woody tissues, identifying the insect vector, or finding traces of the fungal mycelium
on the beetle vector. Lamarche *et al*. [37] tested, in qPCR assays, the effectiveness of the detec-
tion of numerous forest pathogens foreign to the Canadian forest environment, including *G.
morbida*. Oren *et al*. [24] developed a method to identify *G. morbida* and *P. juglandis* using the
species-specific GS 004 microsatellite locus. This method proved effective, but also fairly elabo-
rate and time-consuming, as well as with a sensitivity lower than that provided by a qPCR
probe. Moore *et al.'s* [15] molecular assay, instead, applied conventional PCR for detecting *G.
morbida* on various insect species; this assay, carried out with the *G. morbida*-specific primers
GmF3 and GmR13 (targeting the beta-tubulin gene), resulted in 86% accuracy for *G. morbida*
detection. These authors reported the use of the GmF3 and GmR13 primer pair as an improve-
ment in TCD diagnostics compared to previously published protocols; however, end-point
PCR is notoriously less sensitive than the qPCR probe utilised in this study [38]. However,
despite the effort of research to find ever faster and more performing molecular methods, reg-
ulatory agencies still consider sequencing essential for confirming the identity of newly-intro-
duced organisms in previously uncontaminated areas [17]. Consequently, in the present study,
molecular identification of *P. juglandis* was also performed by sequence analysis of a portion
of the COI gene [20, 25], by employing the combination of primers LCO1490 and HCO2198,
which amplify a roughly 710-bp segment in a number of invertebrate species [28].

Despite the short time since the discovery of this fungal/beetle complex, epidemiological
studies into TCD have proliferated in North America, ranging from susceptibility testing of
hosts within the *Juglandaceae* (members in the genera *Juglans* and *Pterocarya*), to exploring
actual and potential insect vectors, and to TCD diagnostics [15–17]. On the European side, on
the contrary, TCD is a relatively recent and almost unknown phytosanitary problem. The dis-
ease, discovered in the EPPO region just a few years ago, is for the moment confined to Italy,
the only foothold of TCD outside of North America. Here, the numerous, worrying sites, scat-
tered throughout the north and, more recently, the center of the country, suggest the disease is
rapidly expanding [19, 25]. In spite of this alarming situation, studies on the bio-ecology, diag-
nosis, population dynamics, adaptation, as well as the management, of these two invasive alien
species in the EPPO region are scarce.

To face this unprecedented threat to walnut natural stands and plantations, we developed here an early detection tool that could help in preventing the further establishment and spread of TCD organisms into disease-free areas of the European territory. Our qPCR probe (simplex and duplex) assays demonstrated specificity and robustness, while dramatically shortening the time of identification of both TCD organisms, though it is somewhat expensive. In fact, being able to detect the fungus from woody tissues, from bark beetles as well as from insect frass, is of fundamental importance for managing this non-indigenous pest complex in the EPPO region. This protocol could be successfully applied in many areas, such as: in checking import-export material, in surveillance of the territory for new phytosanitary disease outbreaks, and in delineation of areas to be quarantined. It could also help improve pest risk analyses (PRAs) on which to base the implementation of regulations restricting the trade in walnut and timber propagating material. Furthermore, phytosanitary inspection services could use this diagnostic tool to trace the entry paths of the insect/fungus complex, in order to track incoming material.

## Supporting information

**S1 Fig. Gene amplification with qPCR probe duplex for *G. morbida* (bleu) and *P. juglandis* (green) from infected woody tissues and adult insects, respectively.** Curves represent different samples for each target. Curves represent different samples for each target.
(TIF)

**S2 Fig. Serial dilutions 1: 5 –qPCR Probe for *G. morbida* and *P. juglandis* with indication of the relative Ct and LOD.**
(PPTX)

## Acknowledgments

The authors would like to acknowledge the personnel of the Plant Protection Services of Tuscany for their support during the field work.

## Author Contributions

**Conceptualization:** Tiziana Panzavolta, Salvatore Moricca.

**Data curation:** Domenico Rizzo, Daniele Da Lio, Linda Bartolini, Giovanni Cappellini, Tommaso Bruscoli, Matteo Bracalini, Alessandra Benigno, Chiara Salemi, Dalia Del Nista, Antonio Aronadio, Tiziana Panzavolta, Salvatore Moricca.

**Formal analysis:** Domenico Rizzo, Linda Bartolini, Giovanni Cappellini, Tommaso Bruscoli, Matteo Bracalini, Alessandra Benigno, Chiara Salemi, Dalia Del Nista, Antonio Aronadio, Tiziana Panzavolta, Salvatore Moricca.

**Investigation:** Domenico Rizzo, Daniele Da Lio.

**Methodology:** Domenico Rizzo, Daniele Da Lio.

**Supervision:** Domenico Rizzo.

**Writing – original draft:** Domenico Rizzo, Tiziana Panzavolta, Salvatore Moricca.

**Writing – review & editing:** Tiziana Panzavolta, Salvatore Moricca.

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
