## [Decision Letter · Decision Letter 0]

2 Oct 2020

PONE-D-20-27061

A duplex real-time PCR with probe for simultaneous detection of *Geosmithia morbida*and its vector *Pityophthorus juglandis*

PLOS ONE

Dear Dr. Panzavolta

Thank you for submitting your manuscript to PLOS ONE. After careful consideration, we feel that it has merit but does not fully meet PLOS ONE’s publication criteria as it currently stands. Therefore, we invite you to submit a revised version of the manuscript that addresses the points raised during the review process.

We look forward to receiving your revised manuscript.

Kind regards,

Andrea Luvisi

Academic Editor

PLOS ONE

Journal Requirements:

2. Please amend either the title on the online submission form (via Edit Submission) or the title in the manuscript so that they are identical.

Reviewers' comments:

Reviewer's Responses to Questions

**Comments to the Author**

1. Is the manuscript technically sound, and do the data support the conclusions?

Reviewer #1: Yes

Reviewer #2: Yes

2. Has the statistical analysis been performed appropriately and rigorously? 

Reviewer #1: Yes

Reviewer #2: N/A

3. Have the authors made all data underlying the findings in their manuscript fully available?

Reviewer #1: Yes

Reviewer #2: Yes

4. Is the manuscript presented in an intelligible fashion and written in standard English?

Reviewer #1: Yes

Reviewer #2: Yes

5. Review Comments to the Author

Reviewer #1: The paper entitled: " A duplex real-time PCR with probe for simultaneous detection of Geosmithia morbida and its vector Pityophthorus juglandis" describes the development of an effective and sensitive molecular tool based on simplex/duplex qPCR assays for the specific detection of the fungal pathogen and its bark-beetle vector. Notably, this new approach could overcome the difficulties inherent in traditional identification and determine an early and rapid detection of TCD organisms of walnut plantations.

The study presents the results of original research, is well presented and structured. All the experiments have been carried out properly, and the data analyzed and interpreted and described in sufficient detail.

Considering these premises, I recommend the paper for publication.

Specific comments:

1. Although the article is easy to read, scrolling, and informative, I suggest to authors to insert a flowchart in the "DNA extraction" section to better highlight the differences in the DNA extraction stages from the four different types of matrices.

2. Line 156, 180, 252, 424, 445 and elsewhere: there are double spaces between words. Please check.

Reviewer #2: Article needs Minor revision, I tracked changes through "Track changes" and please refer to comments. Please reach me if you have any question. Once authors incorporate the changes, the manuscript will be accepted for publication. Thanks

6. PLOS authors have the option to publish the peer review history of their article (what does this mean?). If published, this will include your full peer review and any attached files.

Reviewer #1: No

Reviewer #2: No

---

## [Editor Report · Decision Letter 1]

9 Oct 2020

A duplex real-time PCR with probe for simultaneous detection of *Geosmithia morbida*and its vector *Pityophthorus juglandis*

PONE-D-20-27061R1

Dear Dr. Panzavolta,

We’re pleased to inform you that your manuscript has been judged scientifically suitable for publication and will be formally accepted for publication once it meets all outstanding technical requirements.

Kind regards,

Andrea Luvisi

Academic Editor

PLOS ONE
---

## [Editor Report · Acceptance letter]

14 Oct 2020

PONE-D-20-27061R1 

A duplex real-time PCR with probe for simultaneous detection of *Geosmithia morbida * and its vector *Pityophthorus juglandis*

Dear Dr. Panzavolta:

I'm pleased to inform you that your manuscript has been deemed suitable for publication in PLOS ONE. Congratulations! Your manuscript is now with our production department. 

Kind regards, 

on behalf of

Dr. Andrea Luvisi 

Academic Editor

PLOS ONE